# Genetic Authentication of the Medicinal Plant *Portulaca oleracea* Using a Quick, Precise, and Sensitive Isothermal DNA Amplification Assay

**DOI:** 10.3390/ijms241310730

**Published:** 2023-06-27

**Authors:** Mo-Rong Xu, Fang-Chun Sun, Bo-Cheng Yang, Hsi-Jien Chen, Chia-Hsin Lin, Jai-Hong Cheng, Meng-Shiou Lee

**Affiliations:** 1Department of Chinese Pharmaceutical Science and Chinese Medicine Resources, China Medical University, Taichung 40402, Taiwan; jolly0427@gmail.com (M.-R.X.); victor.yang_0910@outlook.com (B.-C.Y.); jocelyn0625@hotmail.com (C.-H.L.); 2Department of Medicinal Botanicals and Foods on Health Applications, Da-Yeh University, Changhua 515006, Taiwan; fcsun@mail.dyu.edu.tw; 3Department of Safety, Health and Environmental Engineering, Ming Chi University of Technology, New Taipei City 24301, Taiwan; hjchen@mail.mcut.edu.tw; 4Center for Shockwave Medicine and Tissue Engineering, Kaohsiung Chang Gung Memorial Hospital, Chang Gung University College of Medicine, Kaohsiung 833, Taiwan; cjh1106@cgmh.org.tw

**Keywords:** *Portulaca oleracea*, medicinal crop, loop-mediated isothermal amplification (LAMP), traditional Chinese medicine (TCM), authentication

## Abstract

*Portulaca oleracea* (PO) is a commonly known medicinal crop that is an important ingredient for traditional Chinese medicine (TCM) due to its use as a vegetable in the diet. PO has been recorded to be frequently adulterated by other related species in the market of herbal plants, distorting the PO plant identity. Thus, identification of the botanical origin of PO is a crucial step before pharmaceutical or functional food application. In this research, a quick assay named “loop-mediated isothermal amplification (LAMP)” was built for the specific and sensitive authentication of PO DNA. On the basis of the divergences in the internal transcribed spacer 2 (ITS2) sequence between PO and its adulterant species, the LAMP primers were designed and verified their specificity, sensitivity, and application for the PO DNA authentication. The detection limit of the LAMP assay for PO DNA identification specifically was 100 fg under isothermal conditions at 63 °C for 30 min. In addition, different heat-processed PO samples can be applied for use in PO authentication in the LAMP assay. These samples of PO were more susceptible to the effect of steaming in authentication by PCR than boiling and drying treatment. Furthermore, commercial PO samples pursued from herbal markets were used to display their applicability of the developed LAMP analysis for PO postharvest authentication, and the investigation found that approximately 68.4% of PO specimens in the marketplace of herbal remedies were adulterated. In summary, the specific, sensitive, and rapid LAMP assay for PO authentication was first successfully developed herein, and its practical application for the inspection of adulteration in PO samples from the herbal market was shown. This LAMP assay created in this study will be useful to authenticate the botanical origin of PO and its commercial products.

## 1. Introduction

*Portulaca oleracea* (PO), generally recognized as purslane, is classified into Portulacaceae and is popularly distributed worldwide for use as a vegetable and medicinal crop. Regarding dietary nutrition, PO contains rich omega-3 polyunsaturated fatty acids and various vitamins and minerals; thus, PO is frequently utilized in mixed salad or broths as a functional potherb [1,2]. PO also has exhibited numerous pharmacological activities, including free radical scavenger, antineoplastic, COX-inhibitor, antimicrobial, and immunoregulatory activities [3]. Presently, the increased demand for PO plants has become prevalent due to wide interest in and acceptance of complementary and alternative medicine (CAM) in countries around the world [4,5]. As a result, the authentication of PO medicinal plants is required to confirm their effectiveness and safety [6,7]. For example, several *Portulaca* species and *Bacopa monnieri* (BM) demonstrate similar morphological features between POs [7]. These PO-related herbs are all recognized by the same Chinese name, “Ma-Chi-Xian”, in folk herbal medicine in Taiwan [6]. These herbs have exhibited the possibility of being used as adulterants to distort the identity of PO plants [7]. In particular, BM has been investigated and proven to be the most abundant adulterant: approximately 50–70% is purposely adulterated in the herbal market; thus, BM is also called “fake purslane” [6,7].

Notably, in the official herbal pharmacopeia of traditional Chinese medicine (TCM), PO is solely authenticated as having the correct botanical origin for use and is habitually applied for cold blood and the treatment of dysentery, as well as infections [8]. Based on the above applications in the diet and pharmaceutical industry, PO is considered a substance with homologous use as medicine and food. Moreover, the PO plant has been acknowledged as a “vegetable of long life” in Chinese folklore according to its multiple characteristics and medicinal and nutritional values [8,9,10].

To date, direct analytical approaches such as macroscopic examination, microscopic examination, and thin-layer chromatographic analysis have been applied for PO authentication [6,8]. However, the consequence of authentication by the above methods might be interference by some restrictions, such as the familiarity of the operator with tissue transection, subjective judgment of the plant’s macro/microscopic observation, or metabolite variation between different plant habitats [11,12,13]. During recent decades, due to its reliability, sensitivity, and specificity, quality control in the pharmaceutical industry has been performed using different developed DNA molecular techniques as an alternative technique for confirming the authenticity of herbs [14,15,16]. The most plentiful established techniques are PCR-based methods, such as PCR-RFLP, RAPD, AFLP, and high-resolution melting (HRM) analysis, which have been demonstrated to be effective for the certification of numerous herbal crops [7,15,16]. Over recent years, a peculiar and well-characterized method for DNA amplification named “loop-mediated isothermal amplification (LAMP)” has been extensively applied for verifying species identities [17,18,19]. The LAMP method, with rapidity, specificity, and sensitivity, is superior to PCR for the diagnosis of the existence of specific DNA [19]. In principle, LAMP requires four or six primers and Bst DNA polymerase to amplify DNA efficiently under isothermal conditions during identification [20]. Bst DNA polymerase has strand-displacement activity; thus, a programmed thermal cycler is not needed to heat the temperature sufficiently high for DNA denaturation. In addition, specific LAMP products with ladder-like DNA patterns can be detected by fluorescent dyeing or DNA electrophoresis. Based on the above-mentioned features and convenience, the LAMP assay is very suitable and offers the potential for the development of an on-site diagnosis/authentication kit for DNA detection [19].

In this work, DNA authentication of PO by a fast and species-specific LAMP assay was developed. Derived from the internal transcribed spacer 2 (ITS2) loci sequence variation between PO and its related adulterants, LAMP primers were designed to evaluate their specificity, sensitivity, and reactivity during DNA amplification for PO authentication. In addition, different heat-treating methods to process PO plants were likewise evaluated by the LAMP assay to assess its applicability for the authentication of PO on the market. Finally, commercial PO samples were practically tested for authentication by this validated LAMP analysis. To the best of our knowledge, this created LAMP assay is the first report demonstrating PO botanical origin identification for authentic application on herbs.

## 2. Results

### 2.1. ITS2 Sequence Alignment and LAMP Primer Design

ITS2 has been recognized as a potential DNA barcode for *Portulaca oleracea* (PO) authentication [7]. To build a quick method for the PO authentication with specificity, ITS2 sequences of PO obtained from GenBank were recalled for sequence alignment and applied to design LAMP primers. As illustrated in Figure 1, the aligned ITS2 sequences of PO and other related specimens, PU, PQ, PPS, and PPI, demonstrated that the discrepancy of ITS2 sequence between PO and those adulterant specimens varied from 8% to 12%. In particular, the variation of ITS2 sequences between BM and PO reached 30%. Based on the above variations in ITS2 sequence between PO and other adulterant specimens, a total of three sets of LAMP primers (PO01, PO02, and PO03) were designed herein with the assistance of the software PrimerExplorer V5 for use in the validation of primer. The sequences of three LAMP primer sets are listed in Table 1 and Appendix A, respectively.

### 2.2. Establishment of a LAMP Assay with Specificity for the PO DNA Identification

The sequences of the PO02 LAMP primer set illustrated their respective annealing positions on the ITS2 template DNA of PO, as depicted in Figure 1. When the LAMP assay was achieved by using the PO02 primer set, a representative DNA ladder-like configuration of the LAMP product was synthesized and showed in the electrophoresis gel if the PO template DNA was present in the test sample (Figure 2A, Lane 1). In the sample containing nontarget DNA, such as PU, PQ, PPS, PPI, and BM, the LAMP product could not be produced by using the PO02 primer set for the LAMP assay (Figure 2A, Lanes 2 to 6). Regarding the PO01 and PO03 primer sets (Appendix A), in contrast, neither primer set specifically amplified the amplicon of PO exclusively (Appendix A). In addition, after evaluating the various conditions for the initiation of the LAMP assay, a reaction temperature of 63 °C was used to isothermally produce the LAMP product with higher DNA intensity than at other temperatures (Figure 2B) and more than 50 min was needed for the LAMP reaction (Figure 2C). The sequence of the LAMP product, including the BstNI restriction enzyme cutting site, was originally replicated from the PO ITS2 amplicon (Figure 1); thus, LAMP products can be specifically digested and deconstructed by BstNI to produce DNA banding with small fragments (Figure 2D, Lanes 1 and 3). Taken together, the above-mentioned consequences reflected that this PO02 LAMP primer set was able to use specifically for rapidly authenticating ITS2 DNA of PO.

### 2.3. Sensitive Authentication of PO DNA by LAMP

To evaluate the detection sensitivity by LAMP, distinct amounts of PO genomic were DNA prepared for use as templates to conduct the reaction. The results exhibited that 10 pg of PO genomic DNA was minimal demanded during the detection by the PO02 LAMP primer set (Figure 3, Lane 4). In addition, the extra loop primers of LAMP, LF, and LB were also used for the evaluation of sensitivity improvement (Table 1). The results demonstrated that only the LF primer added to the PO02 primer set improved the sensitivity 10-fold with respect to the LAMP assay (Figure 4A). When the PO02 LAMP primer set containing the LF primer was used, the detection limit for PO authentication was 1 pg of PO genomic DNA (Figure 4A). A similar result of sensitivity improvement by the LF primer was also demonstrated in the treatment with the addition of LF and LB in the LAMP assay (Figure 4C). However, only the LB primer used in the reaction did not adjust the detection sensitivity significantly (Figure 4B). It is worth noting that the addition of the chemical agent DMSO did not significantly improve the sensitivity of LAMP (Figure 4D–F). However, 4% DMSO addition increased the DNA intensity of the LAMP products (Figure 3 and Figure 4E). In particular, the combination of LF and 4% DMSO addition applied to the LAMP assay was found to significantly ameliorate the detection limit to 100 fg for the authentication of PO DNA (Figure 5A). Moreover, this combination was able to shorten the reaction time required to perform the LAMP assay to within 30 min (Figure 5B).

### 2.4. Heat-Processed PO Authentication Using LAMP

The heating procedure is popularly applied to process herbal materials for TCM production. However, heat processing frequently results in DNA damage. Regarding practical applications, heat-treated PO plant samples were assembled and tested for PO DNA verification with the LAMP analysis. To imitate the heating procedure, three methods, boiling, steaming, and drying, were applied to process the PO sample. As elucidated in Figure 6A, the results demonstrated that the boiled, steamed, and dried PO could be used to extract intact DNA for intensifying the amplicon of PO through the LAMP reaction; thus, LAMP products were clearly exposed in the agarose gel (Figure 6A).

Even in the samples treated with boiling for 30 min, steaming for 30 min, and drying for 7 days, PO DNA was authenticated by the LAMP assay. This manifested that the PO authentication with LAMP was not affected even when it was processed with strict steaming conditions. The assay could still be performed successfully when heat-processed PO was used. In contrast to diagnostic PCR, however, the PO sample, which was steam processed for 30 min, did not exhibit any specific PCR product of 186 bp after the diagnostic PCR assay was performed (Figure 6B). Collectively, heat-handled PO with distinct heating methods, such as boiling, steaming, and drying, did not impact the diagnosis consequences of the PO authenticity by LAMP.

### 2.5. PO Speciments in the Market for Authentication by LAMP

To examine the validated LAMP assay that is competent in investigating the PO authenticity of PO commercial products in the market, nineteen PO samples and their related products gathered from numerous herbal stores were employed to detect the authenticity by LAMP (Figure 7). Apart from the concentrated granule powder of PO (sample No. 19), all eighteen of the PO herbal samples were claimed to be authentic by the herbal seller and demonstrated very similar morphological features (Figure 7E). The results of the LAMP assay are shown in Figure 7A. Detection and confirmation of the authenticities of all PO samples collected from herbal stores by LAMP assay demonstrated that only six of nineteen of the PO samples were of correct botanical origin. Moreover, the results further implied that 68.4% (13/19) of the PO samples gathered from 19 resident herbal suppliers contained adulterants. Thirteen of nineteen PO samples revealed no reaction after the detection by LAMP. Among these LAMP assays that confirmed PO identities, samples 4, 5, 10, 11, and 12 also showed agreement with the conclusions of conventional microscopic examination of tissue transection with respect to PO botanical origin (Appendix A and Table 2). In addition to the LAMP assay for PO authentication, PCR was also used to authenticate the above nineteen PO samples. As illustrated in Figure 7C, only one PO sample, No. 11, of six LAMP-confirmed samples was not detected by PCR, even though the DNA quality of the internal control group was validated (Figure 7D). Based on the above results, the detection rate of LAMP for PO authentication was 100% (6/6), superior to 83.3% (5/6) for PCR. Taken together, the above-established LAMP assay can indeed be practically applied to the herbal market for PO authentication; moreover, the results of the investigation of PO identities also significantly demonstrated rampant adulteration in the herbal market.

## 3. Discussion

In recent decades, herbal materials and their demands have become more prevalent and important because complementary and alternative medicine (CAM) has become broadly accepted and utilized in countries around the world [4,5]. However, in the herbal market, misidentification problems and intentional or accidental adulterations of herbal plants have emerged and developed more seriously [21]. The usage of incorrect herbal materials not only reduces therapeutic efficacy but may perhaps also lead to counter effects on the user’s health. Thus, the authentication of herbal crops and their products has been considered a fundamental and critical step for the pharmaceutical industry when manufacturing herbal products [22].

In this investigation, PO authentication by LAMP using ITS2 as a molecular marker was first successfully developed. The ITS2 sequence variation between PO and PU, PQ, PPS, PPI, and BM ranged from 8% to 30%. These results reflected the great variation of ITS2 regions for LAMP primer designation. Because most of these plants were devised from the same classification of taxonomy, particularly PO, PU, PQ, PPS, and PPI, a lower distant genetic relationship between them and PO was expected. In contrast, the ITS2 sequence of *Bacopa monnieri* (BM), belonging to Plantaginaceae, demonstrated higher sequence variation between PO-related species. The criteria were sufficiently met to further design LAMP primers for PO authentication or the discrimination of adulterants.

During the LAMP reaction using specifically designed LAMP primers, there is no need to perform the DNA denaturation step by heating to a high temperature. Thus, substantial reaction time for DNA amplification is saved. In contrast, PCR requires repeating cycles of heating and cooling for DNA amplification. Practically, this is the main reason why PCR requires more time than LAMP. For the LAMP reaction, primer design is a critical step in performing DNA amplification. However, the complexity of LAMP primer design is higher than that of PCR, even though the software is currently available for primer design assistance. In this work, three LAMP primer sets, PO01, PO02, and PO03, were designed and selected for the evaluation of LAMP reactivity (Table 1 and Appendix A, Figure 1, Appendix A). Significantly, only the PO02 LAMP primer set was successful in performing LAMP. This PO02 primer set can specifically amplify the target DNA to produce LAMP products (Figure 2A and Appendix A). Once the potential primer set was determined, the condition improvement of the LAMP reaction was further fulfilled in the latter step.

This established LAMP assay can be used not only to verify the heat-processed PO samples but also for PO authentication on the herbal market (Figure 6 and Figure 7). Due to the higher sensitivity of the LAMP assay, even very few intact sequences of amplicon DNA in the heat-processed PO samples could be efficiently amplified to produce a specific LAMP product to confirm PO identities with respect to those commercial PO samples. This superiority of the LAMP assay has been similarly exhibited in earlier studies [19,23]. The 100-fold improved sensitivity of LAMP in this work was mainly attributed to the addition of the LF loop primer and DMSO reagent (Figure 5). This combination significantly improves the detection limit from 10 pg to 100 fg. Loop primers principally anneal to the loop structures of the LAMP intermediate product to enhance the DNA amplification of the inner primer during the reaction of LAMP [20,24]. However, the main reason for the improved detection sensitivity of LAMP upon DMSO addition is still not completely elucidated. For PCR, the main possible mechanisms of the increased efficiency of DNA amplification through the use of DMSO might be the decreased Tm value of the primer and the destruction of hydrogen bonds of the DNA structure to prevent secondary structure formation [25,26,27]. Thus, DMSO possibly enables a similar mechanism of action for LAMP to ameliorate the efficiency of nucleic acid amplification and the sensitivity of the LAMP reaction.

During authentication by LAMP, six of nineteen PO samples were detected, and their PO identities were verified. The diagnostic consequences of PO authentication by LAMP exhibit complete agreement with the microscopic examination (Figure 7A and Appendix A). In contrast, the detection rate of PCR is only approximately 83.3%. This obtained result might be attributed to the lower DNA quality: fewer intact amplicons are contained in the samples. Even PCR has a sufficiently high sensitivity for use in PO authentication (Appendix A). Thus, LAMP exhibited higher applicability than PCR during PO authentication in practical usage. It is worth mentioning that conventional PCR and LAMP employ the same diagnostic DNA molecular method for the detection of DNA products after DNA amplification. These methods are only suitable for the authentication of homogeneous samples. Otherwise, the establishment of multiplex LAMP for multispecies identification is required [28]. Even so, LAMP is still a potential tool for high-throughput authentication with high sensitivity, specificity, and time effectiveness.

## 4. Materials and Methods

### 4.1. Sample Collection

Five *Portulaca* species, including *Portulaca oleracea* (PO), *Portulaca umbraticola* (PU), *Portulaca psammotropha* (PPS), *Portulaca pilosa* (PPI), and *Portulaca quadrifida* (PQ), were collected throughout Taiwan. *Bacopa monnieri* (BM) plants in Taiwan were also gathered, and all the above-obtained samples were authenticated by Dr. Wen-Te Chang at the LiFu Museum of Chinese Medicine at China Medical University (CMU) based on taxonomy and DNA sequences and were used as standard herbal samples (Table 3). A total of 19 PO commercial specimens were assembled from the different herbal stores (Table 2). Among them, 18 PO products were purchased from different regions in Taiwan’s northern, central, southern, and outer islands. In addition, one of the 19 PO commercial samples was Chinese medicine granules (CCMGs) purchased from a local herbal store (manufactured by a Chinese medicine factory in Taichung, Taiwan). The appearance of PO herb samples is shown in Figure 7E, and microscopic sections are shown in Appendix A.

### 4.2. DNA Extraction

Each plant sample was milled into fine particles using liquid nitrogen, and then the DNA was purified by the commercial plant DNA isolation kit (DNeasy Plant Mini Kit, QIAGEN, Hilden, Germany) according to the manufacturer’s procedures. The purified DNA was stocked in a freezer at −20 °C, and the DNA concentration was analyzed using a spectrophotometer at 260 nm.

### 4.3. Sequence Alignment and Primer Design for LAMP

The ITS2 DNA sequences of all species used in this study were attained from the GenBank of NCBI (https://www.ncbi.nlm.nih.gov/genbank/, accessed on 26 June 2022), as submitted previously [7]. This sequence of selected species was applied for primer design after sequence alignments. The accession numbers of the ITS2 sequences used for PO, PU, PPS, PQ, BM, and PPI are MZ497386.1, MZ505443.1, MZ520546.1, MZ520547.1, MZ520548.1, and MZ531899.1, respectively. The above DNA sequences of the six species from GenBank were subjected to sequence alignment with MEGA-X and Bio-Edit 7.2 (Tom Hall) software and then further compared employing ClustalW (http://www.ebi.ac.uk/clustalw/). The sequence alignment of ITS2 on the six species is shown in Figure 1. The specific LAMP primers (F3, B3, FIP, BIP, LF, and LB) for PO authentication were designed by PrimerExplorer software V5 (https://primerexplorer.jp/e/) derived from the ITS2 sequence variation between PO and its related adulterants. All designed LAMP primers for the authentication of PO are shown in Table 1.

### 4.4. LAMP

The reaction mixture of LAMP contained 1× ThermoPol Buffer (NEB, Ipswich, MA, USA), 6 mM MgSO4 (NEB, USA), 1.4 mM dNTP Mix (GeneMark, Taipei, Taiwan), 1.6 μM of respective inner FIP and BIP primer, 0.2 μM of respective outer F3 and B3 primer, 8 U of Bst DNA Polymerase (M0275L, NEB, USA), and different amounts of extracted genomic DNA, and the rest was mixed with sterile water to a total volume of 25 μL. After mixing well, the reaction was carried out in a 96-well thermal cycler (Veriti Thermal Cycler, Thermo Fisher Scientific, Waltham, MA, USA). The LAMP conditions were implemented at 63 °C for 60 min and then followed by a heating treatment for 10 min at 80 °C for the inactivation of enzyme catalysis. After the reaction, the resultant LAMP products were stocked at 4 °C up to further analysis. Optionally, to determine the effects of loop primers (LF and LB, in Table 1) and DMSO (dimethyl sulfoxide) treatment on the reactivity of LAMP, 0.4 μM of each loop primer and different concentrations ranging from 2% to 6% DMSO were supplemented to the reaction mixture for assessing LAMP product formation. As to negative and positive control, no DNA was added in the tube, and PCR amplified ITS2 DNA of PO was, respectively, used as template DNA.

### 4.5. Detection of LAMP Products

After the LAMP reaction, each LAMP product with 5 μL was loaded onto a 2% agarose gel (containing 0.01% SYBR green dye (SYBR^TM^ safe DNA gel stain)) to achieve DNA electrophoresis for 25 min, and then the gel was placed on an electrophoresis transilluminator (BLooK™, GeneDireX, Inc., Taoyuan, Taiwan) for observation of the LAMP product. Alternatively, 10 µL of LAMP product was mixed with 1 µL of 1000× diluted SYBR^TM^ safe DNA gel stain (Invitrogen^TM^, Thermo Fisher Scientific, Waltham, MA, USA), and then the stained LAMP product was observed directly by the naked eye under UV light illumination.

### 4.6. Specificity of the LAMP Assay

*Portulaca oleracea* (PO) and other specimens (PU, PPS, PPI, PQ, and BM) used in this work are revealed in Table 1. Ten nanograms of purified genomic DNA from each specimen were employed as the template in the reaction mixture to perform the LAMP assay for evaluating the LAMP primer’s specificity for PO authentication.

### 4.7. Optimal Temperature Determination for the LAMP Assay

To determine the optimal temperature for performing the LAMP assay, different temperatures of reaction varying from 60 to 68 °C were applied in the LAMP assay under the addition of ten nanograms of extracted PO genomic DNA. After the reaction was performed at different reaction temperatures, the resultant products were examined for the production and DNA intensities of the LAMP products. The optimal temperature used in the LAMP assay was the reaction temperature employed if the highest DNA intensity of the LAMP product was observed.

### 4.8. Determination of Reaction Times for LAMP Assay

To determine the minimal time of reaction required for performing the LAMP assay, different reaction times, including 10, 20, 30, 40, 50, and 60 min, were used and examined under the addition of ten nanograms of extracted PO genomic DNA as a template. After the LAMP reaction was set up for different reaction times, each produced resultant product was examined to evaluate the production of the LAMP product.

### 4.9. Digestion of LAMP Product by Restriction Enzyme

To confirm the specificity of the LAMP product, each LAMP product was examined by BstNI (Thermo Scientific Inc., USA) digestion. The mixture for the digestion reaction contained 10× buffer R, 20 U of BstNI, and 2 µL of LAMP product, and the remaining volume was deionized water added to yield an overall volume of 25 μL. The digestion reaction was performed at 37 °C for 4 h. After the reaction, a 5 μL aliquant of the LAMP product was checked through electrophoresis with a 2% agarose gel.

### 4.10. Detection Sensitivity of LAMP

To decide the detection sensitivity of LAMP, the PO genomic DNA with different quantities (1 ng, 100 pg, 10 pg, 1 pg, 100 fg, 10 fg, and 1 fg) were prepared and employed as templates for the LAMP assay. After the LAMP analysis was completed, the minimal DNA used in the assay was determined as the detection limit for initiating the LAMP reaction.

### 4.11. PCR

For the detection of PO DNA from all specimens by species-specific PCR, the onward primer PO02-F (5′-GCTTCGCGTCTCCCCC-3′) and backward primer PO02-R (5′-CCAACAAGCCCATCCCAACA-3′) based on the ITS2 DNA sequence were designed and used for PO authentication. For the 5.8S ribosomal DNA amplification as a reference to all extracted DNA samples, the forward primer 5.8S-F (5′-AACGACTCTCGGCAACGGATA-3′) and reverse primer 5.8S-R (5′-AACTTGCGTTCAAAGACTCG-3′) were used for the synthesis of internal control DNA. The reaction mixture volume used for the PCR was 25 µL, which contained 1× PCR buffer for PowerUp^TM^ SYBR^TM^ Green Master Mix (Thermo Fisher Scientific, USA) and 0.4 µM of each primer. A PCR thermal cycler was employed to perform DNA synthesis using the subsequent cycling conditions: early denaturation at 95 °C for 30 s, DNA denaturation with 35 cycles at 95 °C for 30 s, primer annealing at 58 °C for 10 s, extension of the synthesized DNA at 72 °C for 1 min, and the end of a final extension at 72 °C for 1 min. After the reaction, a 5 μL aliquot of PCR product for each specimen was addressed by electrophoresis with a 2% agarose gel.

### 4.12. Using the LAMP Assay for Authentication of Heat-Processed PO Samples

Fresh PO plants were harvested and processed with three heating methods: boiling, steaming, and drying. For boiling treatment, the plants were subjected to boiling water for heating for 10, 20, and 30 min; for steaming treatment, the plants were sterilized in a high-pressure autoclave at 1.5 atm at 121 °C for 10, 20, and 30 min. With respect to drying, the plants were baked in an oven at 40 °C for 1, 4, and 7 days. The purified genomic DNA of these heat-processed PO specimens was used as the template for LAMP and PCR assays.

## 5. Conclusions

In conclusion, the PO DNA identification by LAMP analysis was first evolved in this exploration. Under isothermal circumstances, PO authentication with high specificity, sensitivity, and quickness was performed by specific DNA amplification. The LAMP assay for PO authentication is very useful as a model for establishing a molecular authentication method for on-site application to the identification of other medicinal crops.

## Figures and Tables

**Figure 1 ijms-24-10730-f001:**
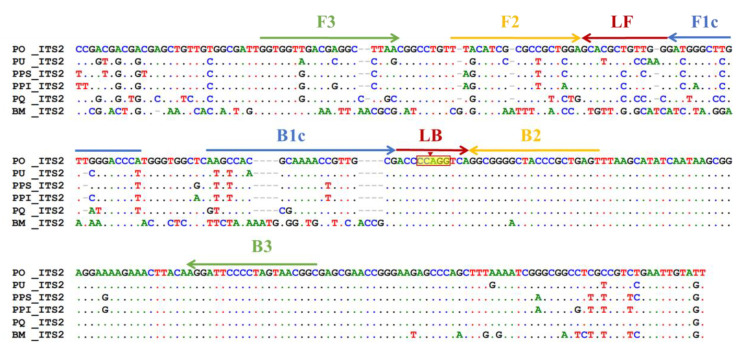
The annealing position for the designed LAMP primer set is depicted in the alignment of the consensus sequence of ITS2. The arrow symbols indicate the direction of DNA polymerization from the PO02 LAMP primer set, which includes outer primers F3 and B3, inner primers FIP (F2 + F1c) and BIP (B2 + B1c), and loop primers LF and LB. Differently colored arrows indicate annealing sites on the different strands. The red box indicates the cutting site of the restriction enzyme BstNI on ITS2 of PO. The hyphen symbols indicate deleted nucleotides in the sequences. Dots with different colors indicate the same nucleotide between sequences.

**Figure 2 ijms-24-10730-f002:**
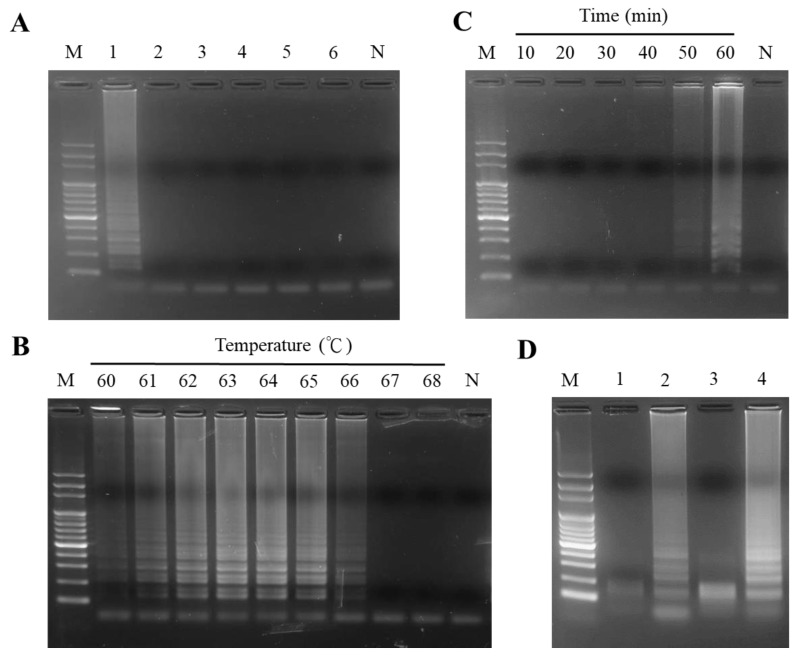
(**A**) The specificity of the LAMP reaction was examined using the LAMP primer set PO02. Lane M, DNA marker; Lanes 1–6 contain the genomic DNA of *P. oleracea* (PO), *P. umbraticola* (PU), *P. psammotropha* (PPS), *P. pilosa* (PPI), *P. quadrifida* (PQ), and *B. monnieri* (BM) used in the LAMP as a template, respectively; Lane N, negative control. (**B**) Effect of various temperatures used in the LAMP assay when using the PO02 primer set. (**C**) Determination of the reaction times of the LAMP assay needed using the PO02 primer set. (**D**) The specificity of LAMP products was analyzed after restriction enzyme digestion. Lane M, DNA marker; Lane 1, LAMP product of PO from Lane 2 digested by BstNI; Lane 2, LAMP product of PO produced by a LAMP assay using PO genomic DNA as a template; Lane 3, LAMP product of PO from Lane 4 digested by BstNI; Lane 4, LAMP product of PO produced by LAMP assay using the ITS2 amplicon of PO as template DNA.

**Figure 3 ijms-24-10730-f003:**
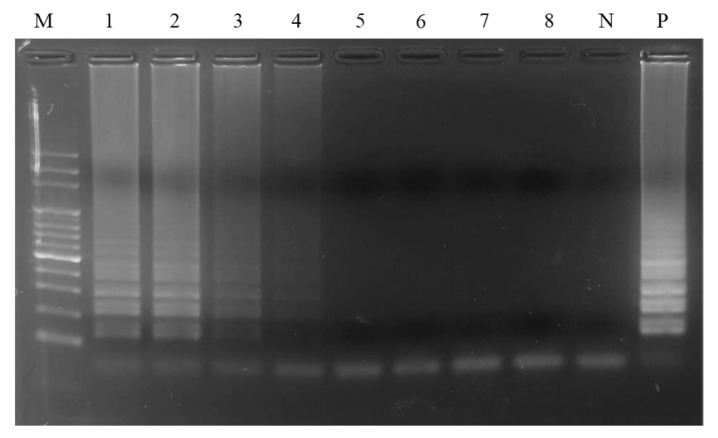
Analysis of the sensitivity of LAMP assay. The LAMP assay was performed by using the PO02 primer set. Lane M, 100 bp DNA ladder marker; Lanes 1–8, various amounts of PO genomic DNA (10 ng, 1 ng, 100 pg, 10 pg, 1 pg, 100 fg, 10 fg, and 1 fg) were used as templates for the LAMP assay; Lane N, negative control; Lane P, positive control.

**Figure 4 ijms-24-10730-f004:**
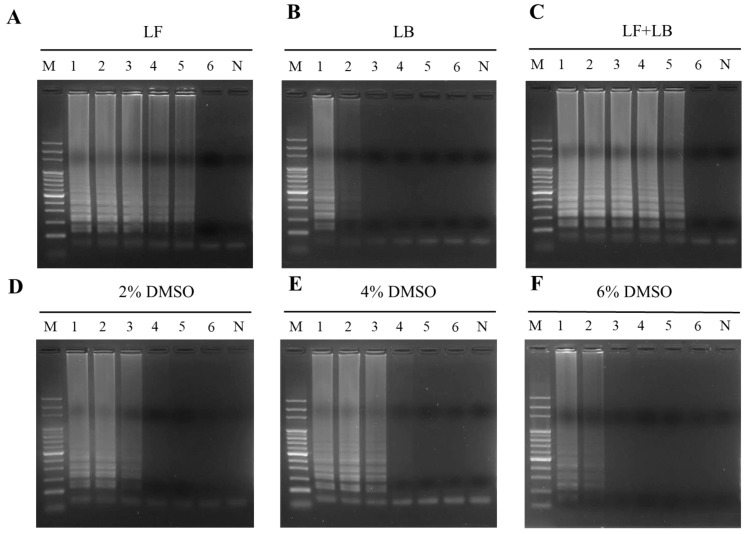
Effects of loop primer and DMSO addition on the sensitivity of the LAMP assay. Loop primers, LB and LF, were individually and simultaneously used in the PO02 primer set for performing the LAMP assay (**A**–**C**). Different amounts of DMSO ranging from 2 to 6% were added for the evaluation of sensitivity improvement in the LAMP assay using the PO02 primer set only (**D**–**F**). Lane M, 100 bp DNA ladder marker; Lanes 1–6 represent 10 ng, 1 ng, 100 pg, 10 pg, 1 pg, and 100 fg of PO genomic DNA, respectively, used as templates for a LAMP assay. Lane N represents the negative control.

**Figure 5 ijms-24-10730-f005:**
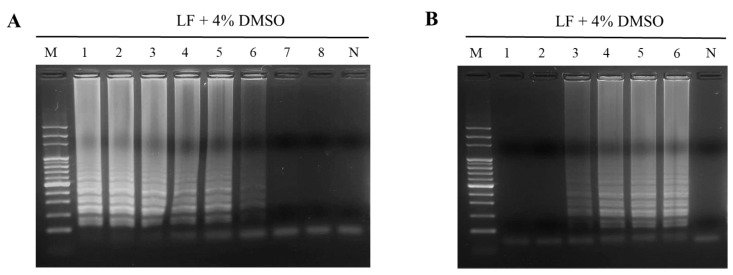
Effect of the combination of LF primer with 4% DMSO addition on the improvement of sensitivity (**A**) and time for reaction (**B**). The LAMP assay was performed using the PO02 primer set combined with the LF primer under 4% DMSO addition; then, the sensitivity and reaction time of LAMP was determined. For Panel (**A**), Lane M indicates a 100 bp DNA ladder marker; Lanes 1–8 represent 10 ng, 1 ng, 100 pg, 10 pg, 1 pg, 100 fg, 10 fg, and 1 fg of PO genomic DNA used respectively for the LAMP assay; Lanes N display negative control. In Panel (**B**), Lanes 1–6 represent 10, 20, 30, 40, 50, and 60 min reaction times used in the LAMP assay. Lane M, 100 bp DNA ladder marker; Lane N, negative control.

**Figure 6 ijms-24-10730-f006:**
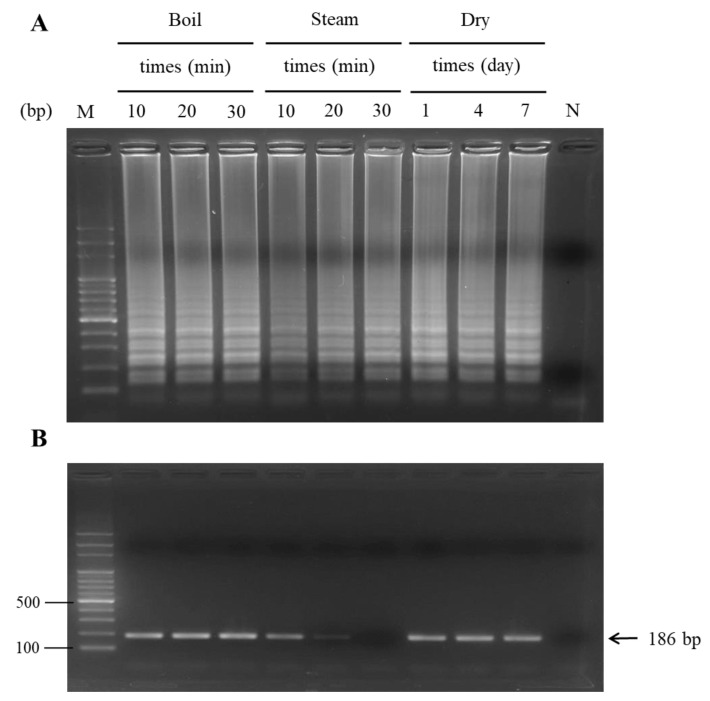
Effect of the PO sample treated with different processing methods on PO authentication by LAMP (**A**) and PCR (**B**) assays. Three processing methods, boiling, steaming, and drying, were used to treat the PO samples. Then, genomic DNA was obtained from processed PO samples for use in LAMP and PCR assays for PO authentication. Lane M, 100 bp DNA ladder marker; Lane N, negative control.

**Figure 7 ijms-24-10730-f007:**
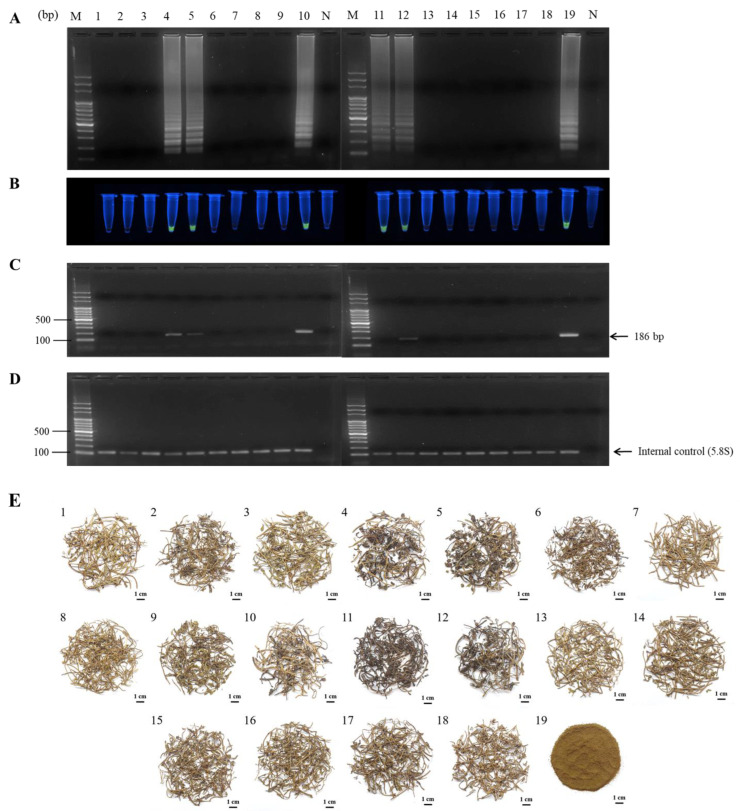
Authenticity analysis by LAMP and PCR on commercial PO samples collected from various herbal markets. Nineteen PO samples, including one sample of PO concentrated granules, were collected from various herbal markets. These samples were used to extract total DNA for use in LAMP (**A**) and PCR (**C**) assays for PO authentication, and all extracted DNA samples were used for 5.8S ribosomal DNA amplification as a reference for the internal control of LAMP and PCR assays (**D**). The fluorescent LAMP product was observed by SYBR green staining under UV excitation (**B**). (**E**) Photographs of the dried PO samples used in the LAMP and PCR assays. Samples No. 1 to 18 represents different commercial PO samples collected from various herbal markets. Sample No. 19 is the PO concentrated granules used in this study. LAMP and PCR assays were performed as described in Section 4. Lane M, 100 bp DNA ladder; Lanes 1–18, PO samples No. 1–18 collected from various herbal stores, respectively; Lane 19, PO concentrated granules “KAISER”; Lane N, negative control.

**Table 1 ijms-24-10730-t001:** LAMP primer used in this study.

Primer Set	Primer Name	Primer Sequence (5′-3′)	Length	Tm	GC%
PO02	F3 ^a^	GGTGGTTGACGAGGCTTAA	19	51.1	53
	B3 ^a^	GCCGTTACTAGGGGAATCCT	20	53.8	55
	FIP (F1c + TTTTT + F2) ^b^	TGGGTCCCAACAAGCCCATCTTTTTTTACATCGCGCCGCTGGA	43	71.2	53
	BIP(B1c + TTTTT + B2) ^b^	AAGCCACGCAAAACCGTTGCGTTTTTACTCAGCGGGTAGCCCCGCC	46	74.3	59
POL	LF ^c^	CCAACAGCGTGC	12	36.2	67
	LB ^c^	ACCCCAGGTCA	11	29.9	64

^a^ Outer primer, ^b^ inner primer, ^c^ loop primer.

**Table 2 ijms-24-10730-t002:** Comparison of identification consequences between LAMP, PCR, and conventional microscopic examination on various commercial dried *Portulaca oleracea* (PO) plants collected from different herbal markets.

No.	Species (Labeled Species)	Location	Characteristics	Identification		
				LAMP	PCR	Microscopical ^a^
1	*Portulaca oleracea*	Yilan	medicinal material	ND	ND	*Bacopa monnieri*
2	*Portulaca oleracea*	Taipei	medicinal material	ND	ND	*Bacopa monnieri*
3	*Portulaca oleracea*	Miaoli	medicinal material	ND	ND	*Bacopa monnieri*
4	*Portulaca oleracea*	Taichung	medicinal material	*Portulaca oleracea*	*Portulaca oleracea*	*Portulaca oleracea*
5	*Portulaca oleracea*	Taichung	medicinal material	*Portulaca oleracea*	*Portulaca oleracea*	*Portulaca oleracea*
6	*Portulaca oleracea*	Taichung	medicinal material	ND	ND	*Bacopa monnieri*
7	*Portulaca oleracea*	Taichung	medicinal material	ND	ND	*Bacopa monnieri*
8	*Portulaca oleracea*	Taichung	medicinal material	ND	ND	*Bacopa monnieri*
9	*Portulaca oleracea*	Taichung	medicinal material	ND	ND	*Bacopa monnieri*
10	*Portulaca oleracea*	Taichung	medicinal material	*Portulaca oleracea*	*Portulaca oleracea*	*Portulaca oleracea*
11	*Portulaca oleracea*	Taichung	medicinal material	*Portulaca oleracea*	ND	*Portulaca oleracea*
12	*Portulaca oleracea*	Taichung	medicinal material	*Portulaca oleracea*	*Portulaca oleracea*	*Portulaca oleracea*
13	*Portulaca oleracea*	Changhua	medicinal material	ND	ND	*Bacopa monnieri*
14	*Portulaca oleracea*	Yunlin	medicinal material	ND	ND	*Bacopa monnieri*
15	*Portulaca oleracea*	Kaohsiung	medicinal material	ND	ND	*Bacopa monnieri*
16	*Portulaca oleracea*	Tainan	medicinal material	ND	ND	*Bacopa monnieri*
17	*Portulaca oleracea*	Pingtung	medicinal material	ND	ND	*Bacopa monnieri*
18	*Portulaca oleracea*	Kinmen	medicinal material	ND	ND	*Bacopa monnieri*
19	*Portulaca oleracea*	Taichung	Concentrated Granules	*Portulaca oleracea*	*Portulaca oleracea*	ND

^a^ Microscopic examinations were performed according to the procedure described in the previous report [6]. These identification results were photographed in Appendix A. ND, not detected.

**Table 3 ijms-24-10730-t003:** *Portulaca oleracea* (PO) and its related adulterant plants used in this study.

Species	Location for Collection in Taiwan	Characteristics	Sample Abbreviation	Accession Number in Genbank
*Portulaca oleracea*	Tainan	whole plant	PO	MZ497386.1
*Portulaca umbraticola*	Tainan	whole plant	PU	MZ505443.1
*Portulaca psammotropha*	Tainan	whole plant	PPS	MZ520546.1
*Portulaca pilosa*	Tainan	whole plant	PPI	MZ531899.1
*Portulaca quadrifida*	Tainan	whole plant	PQ	MZ520547.1
*Bacopa monnieri*	Yilan	whole plant	BM	MZ520548.1

## Data Availability

No dataset was created.

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
