# Peer review of "Genetic Authentication of the Medicinal Plant Portulaca oleracea Using a Quick, Precise, and Sensitive Isothermal DNA Amplification Assay"

_ijms, 2023, doi:10.3390/ijms241310730_

Round 1
Reviewer 1 Report
The reviewed manuscript is dedicated to the design and validation of LAMP-based assay detecting Portulaca oleracea, a valuable crop and medicinal herb. Here, authors designed LAMP test for detection Portulaca oleracea, assessed its sensitivity and specificity. The presented results are timely and interesting for scientists, specializing on the field of molecular diagnostics. However, several issues still need to be addressed before acceptance.
Major issues:
1. Figure 1. Plausibly, positioning of primers in a way to make mismatches right at 3‘-ends of primers would increase the discriminating ability of the designed LAMP. Also, addition of the same figures for other two sets of primers in the supplementary would increase the readability of the manuscript.
2. 4.5. Detection of LAMP Products. It is known that LAMP produces a large number of DNA products which are dangerous in terms of contamination, increasing the probability of false-positive results. Have the authors considered other methods of detections, for example, amplification in PCR strips or tubes with dried SYBR Green I on strip or tube’ caps?
3. Figure 4. Plausibly, the combination of both loop primers improved LAMP efficacy better than the LF primer only, as it can be seen by relatively higher intensity of LAMP products in lanes 4 and 5. Also, loop primers could change the optimal temperature of the assay. Have authors performed temperature gradient with all 6 LAMP primers? It is necessary for proper optimization of LAMP.
4. Page 7, lane 235: “the DNA quality of the internal control group was validated” — authors are encouraged to add information about this PCR in the Material and Methods section.
Minor issues:
1. The language style needs to be polished.
2. Authors are encouraged to mark lines in all panes of Figures 6 and 7.
3. Page 7, line 232: ”Supplementary Fig. 3 and Table 2” — plausibly, Table 3 was meant.
4. Page 9, lines 270-271: “For the beginning of the LAMP reaction, primer design is a critical step to perform and optimize DNA amplification.” — the statement is not clear, authors are encouraged to reformulate it.
5. 4.9. Digestion of LAMP product by restriction enzyme. The paragraph is repeated.
6. 4.10. Detection sensitivity of LAMP — was a carrier used to prevent sorption of DNA on tube walls.
7. Page 13, line 423: “TTo”
Please, find the comments about language quality in the Comments and Suggestions for Authors section.
Author Response
Reviewer 1
Comments and Suggestions for Authors
The reviewed manuscript is dedicated to the design and validation of LAMP-based assay detecting Portulaca oleracea, a valuable crop and medicinal herb. Here, authors designed LAMP test for detection Portulaca oleracea, assessed its sensitivity and specificity. The presented results are timely and interesting for scientists, specializing on the field of molecular diagnostics. However, several issues still need to be addressed before acceptance.
Major issues:
- Figure 1. Plausibly, positioning of primers in a way to make mismatches right at 3‘-ends of primers would increase the discriminating ability of the designed LAMP. Also, addition of the same figures for other two sets of primers in the supplementary would increase the readability of the manuscript.
Responses: We very appreciate reviewer’s comments and suggestion concerning our manuscript. According to reviewer’s comment, we have added other two sets of the annealing position figures of LAMP primers tested in this study into the supplementary file. We believe this revision will be increase the readability of the manuscript.
- 5. Detection of LAMP Products. It is known that LAMP produces a large number of DNA products which are dangerous in terms of contamination, increasing the probability of false-positive results. Have the authors considered other methods of detections, for example, amplification in PCR strips or tubes with dried SYBR Green I on strip or tube’ caps?
Responses: We very appreciate reviewer’s comments and suggestion concerning our manuscript. As to the consideration taken from reviewer, indeed, LAMP assay is very sensitive during the detection is performing. Practically, the false-positive result of LAMP comes from the contamination resulting from operator of test. As to PCR-based assay, it is also quite stable and precise method for the molecular detection. However, different researches based on their vary needs; different molecular assays were developed including LAMP or PCR for their diagnosis. Importantly, no matter what assay was applied for the detection, any kinds of contamination raised from operator or environment should be avoided during performing experiments.
- Figure 4. Plausibly, the combination of both loop primers improved LAMP efficacy better than the LF primer only, as it can be seen by relatively higher intensity of LAMP products in lanes 4 and 5. Also, loop primers could change the optimal temperature of the assay. Have authors performed temperature gradient with all 6 LAMP primers? It is necessary for proper optimization of LAMP.
Responses: We very appreciate reviewer’s comments and suggestion concerning our manuscript. As to the consideration taken from reviewer, indeed, many parameters may influence the reactivity of assay. Basically, LAMP reaction is initiated by Bst DNA polymerase using two inner and two outer primers. In our research group, we have published several diagnosis works based on LAMP assay. In our opinion, these four typical LAMP primers are absolutely needed in the DNA amplification process. Bst DNA polymerase catalyze DNA synthesis under isothermally condition ranged from 60-65℃. Also, Bst DNA polymerase it can tolerant boarder temperature variation between 60-65℃. Designed LAMP primers should be annealed on the template DNA theoretically based on primer’s Tm value under above isothermal condition. As to the loop primer of LAMP, it is alternative for use depending on the improvement of sensitivity. Thus, in conclusion, primer designing is trickier than the temperature of reaction during LAMP is performed. In our case, the temperature gradient we have also done they did not influenced the testing result of sensitivity.
- Page 7, lane 235: “the DNA quality of the internal control group was validated” — authors are encouraged to add information about this PCR in the Material and Methods section.
Responses: We very appreciate reviewer’s comments and suggestion concerning our manuscript. As to the comment taken from reviewer, the PCR information for internal control DNA amplification authors have added in the “Material and Methods (M&M) section”. Please check the added information in the M&M section of“4.11. PCR”.
Minor issues:
- The language style needs to be polished.
Responses: We very appreciate reviewer’s comments and suggestion concerning our manuscript. The manuscript has English polished by American Journal Experts (AJE) before submission. Please check the AJE website (https://secure.aje.com/en/certificate) using the verification code 0852-EDD6-91BD-82FA-688D.
- Authors are encouraged to mark lines in all panes of Figures 6 and 7.
Responses: We very appreciate reviewer’s comments and suggestion concerning our manuscript. Authors have check all line depicted in the figure 6 and 7; all of the important lines were marked. Please check the figure 6 and 7 in the manuscript.
- Page 7, line 232: ”Supplementary Fig. 3 and Table 2” — plausibly, Table 3 was meant.
Responses: We very appreciate reviewer’s comments. The description “Supplementary Fig. 3 and Table 2” has corrected.
- Page 9, lines 270-271: “For the beginning of the LAMP reaction, primer design is a critical step to perform and optimize DNA amplification.” — the statement is not clear, authors are encouraged to reformulate it.
Responses: We very appreciate reviewer’s comments. The improper statement has revised in the revised manuscript. Please check the revised context in the manuscript.
- 9. Digestion of LAMP product by restriction enzyme. The paragraph is repeated.
Responses: We very appreciate reviewer’s comments. Now, we have deleted the repeated paragraph according reviewer’s comment. Please check the revised context in the manuscript.
- 10. Detection sensitivity of LAMP — was a carrier used to prevent sorption of DNA on tube walls.
Responses: We very appreciate reviewer’s comments. For detecting the sensitivity of LAMP assay, all test samples was individually extracted and purified for obtaining genomic DNA. All steps are validated for advoiding DNA contaminations.
- Page 13, line 423: “TTo”
Responses: We very appreciate reviewer’s comments. Now, we have corrected the typo error according reviewer’s comment. Please check the revised context in the manuscript.

Reviewer 2 Report
Xu and co-authors developed LAMP test for species-specific identification of Portulaca oleracea. The herb is used in traditional medicine in East Asia and is supposed to be a remedy for different diseases.
I have no crucial objection to the practical part of the research. However, I think the research is not appropriate for IJMS due to low potential interest. I recommend to apply the paper in a more specialized journal.
Some remarks for the manuscript:
1. Line 53-54 – quite strange reference to the Master studies. It would be better using a journal article for the statement. Further citation of this thesis seems inappropriate.
2. Lines 64-66. Such citation requires accurate reference to WHO statement. Please provide it instead of the used ones.
3. Lines 81-82: The LAMP method with rapidity, specificity and sensitivity is superior to PCR for the diagnosis of the existence of specific DNA [19]. – PCR is “gold standard” method of diagnostic. LAMP is faster than conventional PCR. The statement of LAMP superiority over PCR in specificity and sensitivity is quite controversial. Perhaps, authors meant LAMP in cited reference?
4. Please provide the LAMP sensitivity in molar concentration.
5. Lines 350-351. MZ520548.1 should be put after MZ520546.1
6. Doubling of indexes in the References list
The language of the manuscript is acceptable, but some style corrections are needed.
Author Response
Reviewer 2
Comments and Suggestions for Authors
Xu and co-authors developed LAMP test for species-specific identification of Portulaca oleracea. The herb is used in traditional medicine in East Asia and is supposed to be a remedy for different diseases.
I have no crucial objection to the practical part of the research. However, I think the research is not appropriate for IJMS due to low potential interest. I recommend to apply the paper in a more specialized journal.
Responses: We very appreciate reviewer’s comments. According to the previous published paper in the “Internal Journal of Molecular Sciences” (IJMS), IJMS accept similar research work such as molecular diagnosis, species authentication and so on. Thus, authors believe our research work is suitable for publish in the IJMS; and it is readable for scientific community once this manuscript is accepted and for further publication in the IJMS.
Some remarks for the manuscript:
- Line 53-54 – quite strange reference to the Master studies. It would be better using a journal article for the statement. Further citation of this thesis seems inappropriate.
Responses: We very appreciate reviewer’s comments. According to the instruction for author of “Internal Journal of Molecular Sciences; IJMS”, IJMS accept authors citing their reference list in thesis into the manuscript. If a cited reference (thesis) is described as following the citation style of Journal, it would be allowed for use. Thus, authors prefer to keep this citation in the reference list.
- Lines 64-66. Such citation requires accurate reference to WHO statement. Please provide it instead of the used ones.
Responses: We very appreciate reviewer’s comments. Now, we have corrected the improper statement according reviewer’s comment. Please check the revised context in the manuscript.
- Lines 81-82: The LAMP method with rapidity, specificity and sensitivity is superior to PCR for the diagnosis of the existence of specific DNA [19]. – PCR is “gold standard” method of diagnostic. LAMP is faster than conventional PCR. The statement of LAMP superiority over PCR in specificity and sensitivity is quite controversial. Perhaps, authors meant LAMP in cited reference?
Responses: We very appreciate reviewer’s comments. Indeed, in many research cases, PCR is “gold standard” method of diagnostic. Also, PCR is faster, high specific and sensitive. However, in other research cases, LAMP having above advantages is superior over PCR has been also reported. Authors just has cites their statement from reference No. 19.
- Please provide the LAMP sensitivity in molar concentration.
Responses: We very appreciate reviewer’s comments. Currently, the unit of DNA quantity was expressed in several styles such as molarity, weight or copy number. However, the expression of DNA quantity in molarity or in copy number it needs to obtain more information such as the molecular weight of plant’s genomic DNA. Actually, it is impossible to provide the molecular weight of whole genome of plant. This is the main reason that most published papers they has done the sensitivity test using weight (microgram) of template DNA as the unit of DNA quantity.
- Lines 350-351. MZ520548.1 should be put after MZ520546.1
Responses: We very appreciate reviewer’s comments. Now, we have corrected the improper statement according reviewer’s comment. The accession number, MZ520548.1 is expressed after MZ520546.1.
- Doubling of indexes in the References list
Responses: We very appreciate reviewer’s comments. Now, we have corrected the improper indexes according reviewer’s comment.

Round 2
Reviewer 1 Report
Many thanks to authors for their efforts in editing the manuscript. All questions from the first review were answered and all issues solved. The manuscript can be accepted in the present from without further correction.
Reviewer 2 Report
I still consider the reference to the master thesis as inappropriate. However, if the journal policy admits this, I will not oppose it. The same situation with the overall subject of the research. Other remarks have been corrected.